# The Dielectric Behavior of Protected HKUST-1

**Simona Sorbara [1], Nicola Casati [2], Valentina Colombo [3,4], Filippo Bossola [4] and Piero Macchi [1,*]**

[1] Department of Chemistry, Materials and Chemical Engineering, Polytechnic of Milano, Via Mancinelli 7, 20131 Milano, Italy; simona.sorbara@polimi.it

[2] Laboratory for Synchrotron Radiation—Condensed Matter, Paul Scherrer Institute, Forschungstrasse 111, 5232 Villigen, Switzerland; nicola.casati@psi.ch

[3] Department of Chemistry, University of Milan, Via Golgi 19, 20133 Milano, Italy; valentina.colombno@unimi.it

[4] CNR-Institute of Chemical Sciences and Technologies (SCITEC) "Giulio Natta", Via Golgi 19, 20133 Milano, Italy; filippo.bossola@scitec.cnr.it

* Correspondence: piero.macchi@polimi.it; Tel.: +39-0223993023

**Abstract:** We investigated the adsorption properties and the dielectric behavior of a very well-known metal-organic framework (MOF), namely $Cu_3(BTC)_2$ (known as HKUST-1; BTC = 1,3,5-benzenetricarboxylate), before and after protection with some amines. This treatment has the purpose of reducing the inherent hygroscopic nature of HKUST-1, which is a serious drawback in its application of as low-dielectric-constant (low-κ) material. Moreover, we investigated the structure of HKUST-1 under a strong electric field, confirming the robustness of the framework. Even under dielectric perturbation, the water molecules adsorbed by the MOF remained almost invisible to X-ray diffraction, apart from those directly bound to the metal ions. However, the replacement of $H_2O$ with a more visible guest molecule such as $CH_2Br_2$ made the cavity that traps the guest more visible. Finally, in this work we demonstrate that impedance spectroscopy is a valuable tool for identifying water sorption in porous materials, providing information that is complementary to that of adsorption isotherms.

**Keywords:** metal-organic frameworks; dielectric constant; adsorption properties

## 1. Introduction

Within the large array of applications of metal-organic frameworks (MOFs), ever-growing attention is being paid to their use as materials for low-dielectric-constant (low-κ) devices, which are fundamental for the miniaturization of integrated circuits in informatics technology [1]. This is one of the main challenges for current research on advanced and disruptive materials. Scientists and engineers try to design and fabricate new types of insulators as alternatives to the traditionally employed fiberglass, a composite of silica with a relatively high dielectric constant (κ = 3.9) [2], which is not adequate for further miniaturization of the devices. This deficiency causes, for example, a large cross-talk effect in microcircuits. For this reason, materials with lower κ are necessary to guarantee the expected advances in performance and to avoid unwanted processes.

The interest in MOFs is motived by their very nature as very stable, tunable, and highly porous solids that are often crystalline. Indeed, these are key features for disruptive ultra-low-κ materials. In particular, the high porosity implies a very low content of matter (hence of electrons) in the overall volume spanned, which makes them ideal for achieving perfect insulation and approaching the dielectric behavior of a vacuum itself (κ = 1), which is of course the lowest limit. A material with at least ca. 50% empty space inside should be able to guarantee κ < 2, which is one of the targets for the new-generation low-κ materials. Many MOFs are indeed known with such an empty volume inside their structures and sufficient framework stability (i.e., rigidity) to preserve it over time and upon mechanical, thermal and chemical perturbation [3–5].

Although the above description addresses MOFs as seamlessly ideal low-κ materials, some pitfalls may affect their application. In fact, most MOFs are hygroscopic. This functionality is currently being exploited for water-sucking materials, designed to fight against drought [6]. However, for applications in micro and nanoelectronics, hygroscopicity is a fundamental defect, because it implies (partial) occupation of the pores with very mobile and highly polarizable water molecules, affecting the extraordinary dielectric behavior highlighted above [7].

Some possible strategies for MOF design could address this problem. The objective is to design a hydrophobic material with the minimal loss of dielectric performance. This could be achieved using directly hydrophobic building blocks or otherwise by post-synthetic modifications of the MOF with the insertion of functional groups that can guarantee the requested hydrophobicity. These strategies, however, do not always lead to the desired material, because perfect hydrophobicity may not be obtained by simply introducing some hydrophobic functional groups into the building blocks (normally in the organic linkers). In fact, the metals may remain sources of attraction for water molecules, unless completely saturated by the coordinated linkers. Moreover, the linkers may be ambivalent, featuring both hydrophobic and hydrophilic sites.

An alternative is protecting the pores of the MOFs, in order to hamper the insertion of water molecules into the framework. This drastically reduces the MOF's capacity to suck water molecules, without affecting the internal structure, or at least with minimal perturbation limited to the surface and a few inner layers. One example is the approach of coating with polydimethylsiloxane proposed by Zhang et al. [8].

Recently, Gao et al. [9] proposed a strategy to fabricate superhydrophobic and superoleophilic MOF composites, obtained from the surface reaction of the activated MOF (i.e., evacuated from the content in the pores) with octadecyl-ammine (OA). The long alkyl chain of OA with low surface energy was grafted onto the surface of some highly porous MOFs, making them water-resistant and endowing the composites with admirable superhydrophobicity. The possible drawbacks for their application as low-κ materials are the polarizability enhancement of the system due to the insertion of the amine itself and the long-term stability, especially in highly humid environments.

In this study, we focused on one of the most studied MOFs, namely, $Cu_3(BTC)_2$ (see Figure 1), also known as HKUST-1 [10], where BTC stands for the deprotonated form of benzene-1,3,5-tricarboxylic acid (it is worth noting that sometimes the linker is reported in the literature as the trimesic acid anion, TMA). As is well-known, HKUST-1 features a large porosity of up to ca. 70% (as calculated with a probe radius of 1.2 Å) and a very high chemical and thermal stability, but it is very hygroscopic. Its hydrophilicity is in the first instance related to the unsaturated square planar Cu(II) ions present in HKUST-1 in the form of a classical copper acetate paddle-wheel geometry (see Figure 1). It is worth noting that this coordination occurs along a Jahn–Teller distortion direction and is not the strongest binding to the metal ion. Nevertheless, the apical vacancy at the Cu(II) ion represents a preferential adsorption site for water molecules, found already in the as-synthetized material. This, however, explains only a small portion of the overall hydrophilicity, which increases through more traditional electrostatic interactions between water molecules and other sites of the linkers. It is important also to mention that the prolonged exposure of the material to moisture promotes the hydrolysis of Cu-O bonds in the paddle wheels, leading to the breakdown of the crystal structure [11,12]. Indeed, another challenge for today's research is to obtain hydrolytic stability in hemilabile MOFs [13], which can be obtained by changing the linker units or using post-synthetic modifications such as the one described below (see Figure 2).

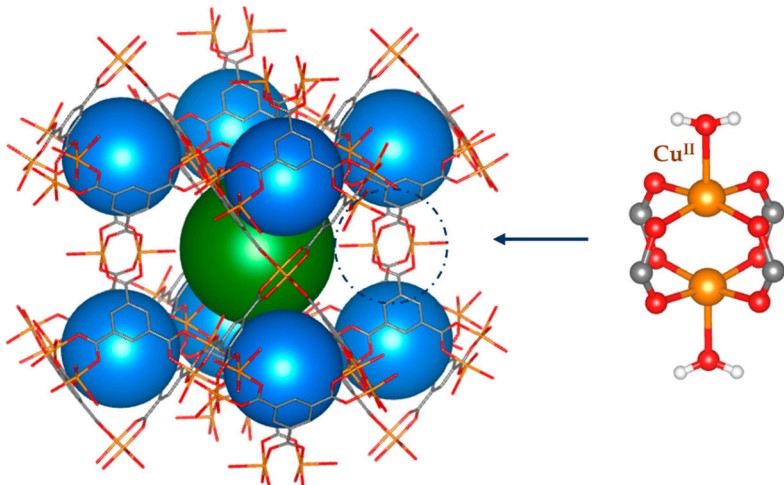

**Figure 1.** The structure of HKUST-1, an example of nanoporous MOFs. The secondary building unit (SBU) is represented on the right.

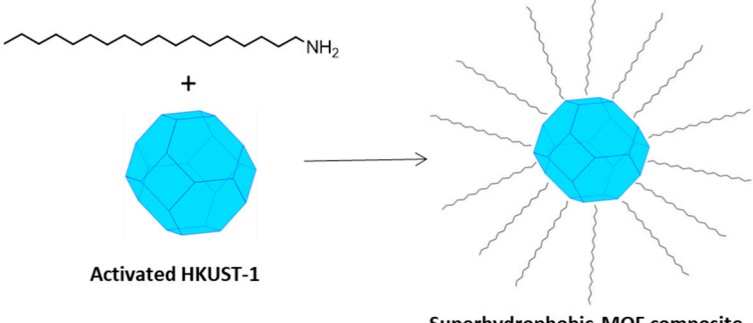

**Figure 2.** A schematic picture of the fabrication of a superhydrophobic MOF composite by means of long-chain alkyl amines (such as OA). This picture is similar to Figure 1 in reference [9].

For these reasons, HKUST-1 represents an ideal candidate to test whether the protection methods described above could be efficiently adopted to make highly porous but hygroscopic MOFs, as suitable low-$\kappa$ materials. In addition, we tested HKUST-1 under the perturbation of an electric field, because its stability under these conditions has not been ascertained so far. Indeed, to date, theoretical calculations have been reported on some MOFs' crystal structures under electric fields, but experiments have not been reported. For example, Ghoufi et al. [14] studied a flexible MOF, MIL-53, that unfortunately cannot be obtained in single-crystal form.

The experiments on dielectric constant and water adsorption that we report here are supported by theoretical calculations of the polarizabilities of the species (building blocks of HKUST-1 and amines used for protection), in order to gain insight into the electronic structure of the material under the perturbation of an electric field and provide useful parameters for the efficient design of new materials based on this approach.

## 2. Materials and Methods

### 2.1. Synthesis

The hydrated form of $Cu_3(BTC)_2$ (HKUST-1) was prepared according to the reported method [15]. Microcrystalline pellets of HKUST-1 were activated by heating to 200 °C for 20 h at $10^{-2}$ mbar. As reported previously by Schlichte et al. [16], this treatment is sufficient for complete removal of all guest molecules.

For better visualization of the guest molecules inside HKUST-1, water was exchanged with $CH_2Br_2$. For this purpose, a HKUST-1 sample was activated in vacuum at high temperature, and 100 mg of the sample was plunged into 5 mL of $CH_2Br_2$.

## 2.2. MOF Surface Protection

The surface reactions of the activated HKUST-1 with different amines were carried out following the prescription of Gao et al. [9] for octadecyl amine. A solution of toluene with the activated MOF and the amine (with a concentration of 10 mM) was stirred at 120 °C for 24 h under nitrogen. In Table 1, we report the quantities adopted for each reaction and the corresponding color observed for the powder of the composite material.

**Table 1.** The amines used for protection of HKUST-1.

| RNH$_2$ | Acronym | Amount of RNH$_2$ (g) | Amount of HKUST-1 (g) | Composite Color |
|---|---|---|---|---|
| Octadecylamine | OA | 1.348 | 1.000 | Dark Blue |
| Decylamine | DA | 0.786 | 1.000 | Dark Blue |
| Amylamine | AM | 0.435 | 1.000 | Dark Blue |
| 1-Naphthylamine | 1NTA | 0.716 | 1.000 | Black |
| Aniline | AN | 0.465 | 1.000 | Dark Green |
| 3-Phenyl-1-propylamine | 3P1PA | 0.676 | 1.000 | Dark Blue |

## 2.3. Dielectric Constant Measurement

The dielectric constant was determined via impedance spectroscopy. The measurements were carried out with a Solartron ModulabXM impedance/gain-phase analyzer equipped with XM MFRA 1 MHz and XM MAT 1 MHz control modules. A 12962A sample holder was used with dried powder pellet samples with a diameter of 13 mm, prepared by applying a force of ca. 15 kN (corresponding to a pressure of ca. 0.1 GPa). The sample holder consists of a two-brass-parallel-electrodes capacitor provided with a guard ring, which reduces the fringing effect of stray fields at the edge of the tested materials. The measurement parameters were controlled with the ModulabXM software. The measurement setup consisted of a fixed-mode-generator voltage level of 0 V with an amplitude of 100 mV and a frequency sweep from 1 Hz to 1 MHz. The electronic field was applied between the two electrodes, across a measured thickness of the sample, at room temperature.

All the pellets activated at 200 °C for 20 h at $10^{-2}$ mbar were loaded into the sample holder in a glovebox under a N$_2$ atmosphere. After the initial measurements at time $t_0$, all the pellets were expose to a stable air humidity of ca. 60% (maintained by storing the samples in a closed box containing humidifier polymers and controlled through a hygrometer) and the dielectric constant was measured at variable intervals: with a frequency of 10 min for the first half-hour, then gradually less frequently up to $t_\infty$ = 13 days after activation. For each sample, we performed at least two measurements, and the reported results are the averaged values.

## 2.4. X-ray Diffraction

Synchrotron X-ray diffraction data were collected at the Material Science beamline of the Swiss Light Source (Paul Scherrer Institute, Villigen, Switzerland) [17]. The system used to collect diffraction data under the electric field consists of two stainless steel electrodes of 0.4 mm diameter and 3.5 mm length. A single-crystal sample (dimensions of ca. 0.1 mm for HKUST-1, ca. 0.08 mm for HKUST-1@CH$_2$Br$_2$) was mounted with silver paste on one of the two electrodes, while the other was placed at a distance of ~1 mm (see Figure 3). The axial system was mounted vertically, and the resulting electric field was applied along this direction, perpendicular to the incoming (horizontal) beam (see Figure 3). The highest applicable voltage was limited to 2 kV due to sparks generated during the diffraction experiments by the ionizing X-rays. For the sake of safety, a potential of 1.5 kV was applied. Because the typical dimensions of the specimens were ~100 μm (linear dimensions), the resulting applied electric field was of the order of magnitude of 0.01 GV/m.

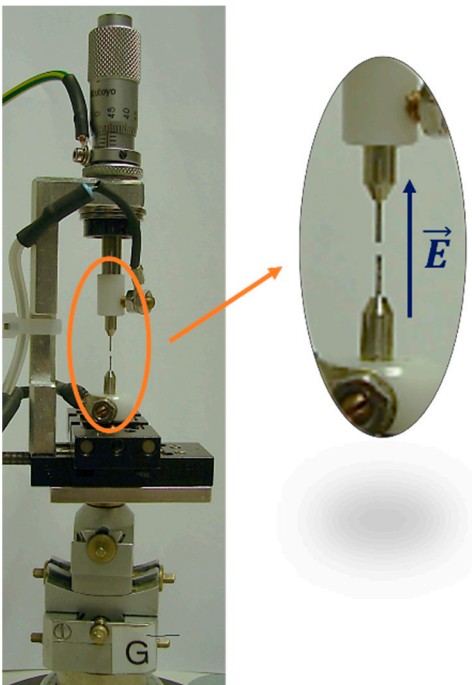

**Figure 3.** The set-up for the measurements under electric field at the Swiss Light Source at the Paul Scherrer Institute.

For HKUST-1 (compound **1**), the diffraction data were collected on the same sample before, during and after the application of the electric field with radiation doses of ca. 25 min separated by 5 min (for the data collection after removing the electric field a double dose was planned, but only the first run was eventually used for the refinements, due to the observed sample decay). For HKUST-1@CH$_2$Br$_2$ (compound **2**), data collection before applying the field was carried out on a different sample, in order to reduce the possible sample decay, and data for second sample were collected during and after the application of the field (for 50 min and 15 min, respectively, after 15 min of relaxation).

A monochromatic beam of radiation with energy ~25.2 keV was adopted, each time calibrated by the diffraction of a Si standard. The 2D diffraction images were recorded with a Dectris Pilatus 6M 2D detector and integrated using the Rigaku-Oxford Diffraction CrysAlis software [18]. Crystal structure refinements were carried out with Olex2 [19]. Mercury software [20] was used to draw the crystal structures and calculate the void volume.

Powder X-ray diffraction was performed on synthesized samples before and after the protection with amines. The data were collected with a Bruker D2 phaser diffractometer, working at 30 kV and 10 mA, using CuK$\alpha$ monochromatized radiation.

### 2.5. Water Absorption Isotherms

Water adsorption isotherms were measured with a Micromeritics ASAP2020 analyzer at 295 K on samples previously degassed for 12 h at 180 °C in high vacuum. The water physisorption properties of three powder samples were evaluated: HKUST-1 to obtain a blank, and 3P1PA and DA, which seem to have the best performances for aromatic and alkyl amines, respectively. In addition, to confirm the different adsorption capacity of a pellet in comparison to a powder, a water adsorption isotherm of a pellet of DA was also obtained.

### 2.6. Theoretical Calculations

Density functional theory (DFT) calculations on all the protecting ammines and their interactions with the HKUST-1 frameworks were performed employing the B3LYP functional [21,22] in combination with the def2TZVP basis set [23,24], using the Gaussian16

software [25] on the HPC supercomputer Galileo-100 of the Italian Cineca computing center. The amine geometries were optimized, and their polarizabilities were calculated by applying couple-perturbed Kohn–Sham theory. The atomic polarizability tensors were calculated via numerical differentiations of the total atomic dipoles, using the PolaBer software [26]. An electric field of 0.0001 atomic units was applied in positive and negative $x$, $y$ and $z$ directions, and the corresponding electron densities were analyzed via the quantum theory of atoms in molecules (QTAIM) [27] using AimAll [28], which calculates atomic dipole moments following the scheme proposed by Bader and Keith [29] and modified by Krawczuk et al. [22]. The inherent asymmetry of the atomic polarizabilities is overcome by the symmetrization scheme of Nye [30].

## 3. Results and Discussion

### 3.1. HKUST-1 under Electric Field

HKUST-1 crystallizes in the cubic space group $Fm\bar{3}m$ and features a high fraction of unoccupied volume, at least ideally, i.e., assuming that all the pores are empty and that all metal ions are not coordinated along the apical site. The three-dimensional periodic structure is generated by the binuclear Cu(II) paddle-wheel secondary building units (SBUs) connected via the tritopic BTC organic linker (see Figure 1). Two kinds of intersecting pores are present with diameters of ca. 10 and 15 Å (see Figure 1). The large (octahedral) cavity in the center of the unit cell (represented in green in Figure 1) is crystallographically equivalent to cavities in the middle of unit cell edges (not shown in the picture) and topologically identical to those at the center of the faces and at the vertexes of the cell (not shown in Figure 1 for sake of simplicity; for more details see Figure S1 in the Supplementary Material). These two cavities correspond to the sites occupied by Na and Cl in NaCl. The smaller (blue) cavity, instead, sits on a tetrahedral site. It is worth noting that the large cavities are directly interconnected through channels along the main crystallographic directions, so that they give rise to uninterrupted empty volumes, whereas the smaller cavities are more closed and connected only to the large cavities through smaller apertures.

After activation, if exposed to a humid atmosphere HKUST-1 rapidly adsorbs water molecules, a phenomenon which is easily observed the change in color from dark blue to cyan. Water molecules can easily access both kinds of pores and even be anchored to the internal surfaces of the MOF. According to a previous study by Scatena et al. [31], we can identify three kinds of water molecules: (a) those freely moving in the pores without any strong interaction with the framework; (b) those connected through hydrogen bonds to the framework linkers; and (c) those weakly coordinated to the metal nodes of the framework along the Jahn–Teller distorted direction (typical for Cu(II) ions). With X-ray diffraction on single crystals, only type (c) water molecules are (partially) visible, because they are quite rigidly constrained to the metals and due to the Cu-Cu bond length increase, while none of the other $H_2O$ molecules are directly visible, and their presence can only be inferred through analysis of the residual electron density inside the channels. As a rigid, second generation [32] type of MOF, the volume increase due to hydration is very small.

Because of the high hydrophilicity, only under special conditions (namely after activation and in an anhydrous atmosphere) can HKUST-1 display the exceptionally low dielectric constant (κ~1.7, see [27]) predicted theoretically and guaranteed by the large voids of the structure, which is the main reason for suggesting MOFs as potentially good low-κ materials.

In the following paragraphs, we discuss methods to protect the {Cu$_3$(BTC)$_2$} framework and obtain the same good dielectric behavior even when an anhydrous atmosphere cannot be guaranteed. In any case, other features are required to elect an MOF such as HKUST-1 as a good low-κ material. For example, HKUST-1 is known for its high thermal and chemical stability, as well as easy synthesis and a tendency to form sufficiently large single crystals of good quality (hence ensuring good reproducibility of the material's structural properties). One additional feature to check is the stability under an electric field, which implies both the breakdown voltage (i.e., the field necessary to break the

insulating behavior) and the structural changes under the field (before the breakdown). Although some studies have reported the incipient conductance of HKUST-1 doped with guest molecules such as tetracyanoquinodimethane [33], to the best of our knowledge the breakdown voltage of pure $\{Cu_3(BTC)_2\}$ has not been established so far. Our interest, in any case, was in the structural stability under an electric field. This study had two purposes. One was to assess the stability of the rigid framework structure under an applied electric field, and the other was to test whether the water molecules inside the MOF channels could be made somewhat visible through X-ray diffraction under an electric field (which would order the guests). Water is indeed very mobile, because it has a small mass and a large dipole, and therefore is easily oriented in an externally applied field. As discussed above, the disorder of water molecules inside the framework channels of HKUST-1 is a drawback, because it means they do not contribute to Bragg diffraction peaks in a measurable way. On the other hand, a reordering of the water molecules inside the channels may result in clearer features being displayed in the diffraction pattern. However, because water does not contain heavy atoms, its contribution to the X-ray diffraction remains small.

The adopted experimental procedures were designed according to the following steps. Diffraction data were collected at ambient temperature, with the electric field initially *off* taken as a benchmark of the unperturbed sample (experiment **1a**), then *on* (experiment **1b**) and eventually *off* again (experiment **1c**) to check effects after the application of a field (see experimental section for more details). To monitor the electric field effects, the structural models were refined using the framework atoms only (including the oxygen atom of the water molecule directly coordinated to Cu), so that the corresponding $F_o$–$F_c$ maps addressed the unassigned electron densities associated with solvent guests inside the pores, apart from experimental errors.

The electron density maps (Figure 4) show the slightly different sizes and shapes of the residuals depending on the applied field, indicating that the external stimulus produces some, albeit very small, effects on the guest water molecules inside the cavities. For example, a small decrease in the residual electron density inside the octahedral cavities is observed when applying a voltage of 1.5 kV, and the largest residual occurs at the octahedral cavity at the center of the cell, evolving from 1.4 to 1.0 $e\text{Å}^{-3}$. Moreover, there seems to be some hysteresis, because these effects persisted even after switching off the electric field.

No significant structural change was observed in the $\{Cu_3(BTC)_2\}$ framework, which confirms the stability of HKUST-1 in a relatively strong electric field, at least well above the realistic operational limits. In fact, the unit cell volume change was below 0.1%, within the typical variance observed in different experiments on the same sample. Moreover, all bond distances and angles within the framework differed by less than $1\sigma$.

In order to make the guest molecules more visible and test how they occupy the pores of the MOF, water was exchanged with another polar solvent containing heavier atoms, namely, dibromomethane (see details in the experimental section). HKUST-1 maintained its crystallinity, which allowed the structure to be refined (isostructural to the hydrated form of HKUST-1) through single-crystal X-ray diffraction. The experiment revealed the $CH_2Br_2$ molecules quite well that seamlessly fitted the tetrahedral cavities and partially occupied them, though of course in a disordered manner (the molecule belongs to a symmetry subgroup of $T_d$). Their occupancy remained stable over time, despite the well-ascertained affinity of HKUST-1 for water. In the larger octahedral cavities, however, there was little evidence of the nature of the content. The residuals at the center of these cavities were larger than for the hydrated samples, which could be due either to better phasing of the reflections due to the Br atoms and/or to some $CH_2Br_2$ molecules being disordered along the channels (but not trapped as for the tetrahedral cavity). The $CH_2Br_2$ stability inside the MOF was evaluated not only by X-ray diffraction but also through bromine X-ray fluorescence, which did not decrease over time, at least judging from the background intensity (mainly due to the fluorescence, considering that the detector was set to measure at an energy threshold below the K-edge of bromine).

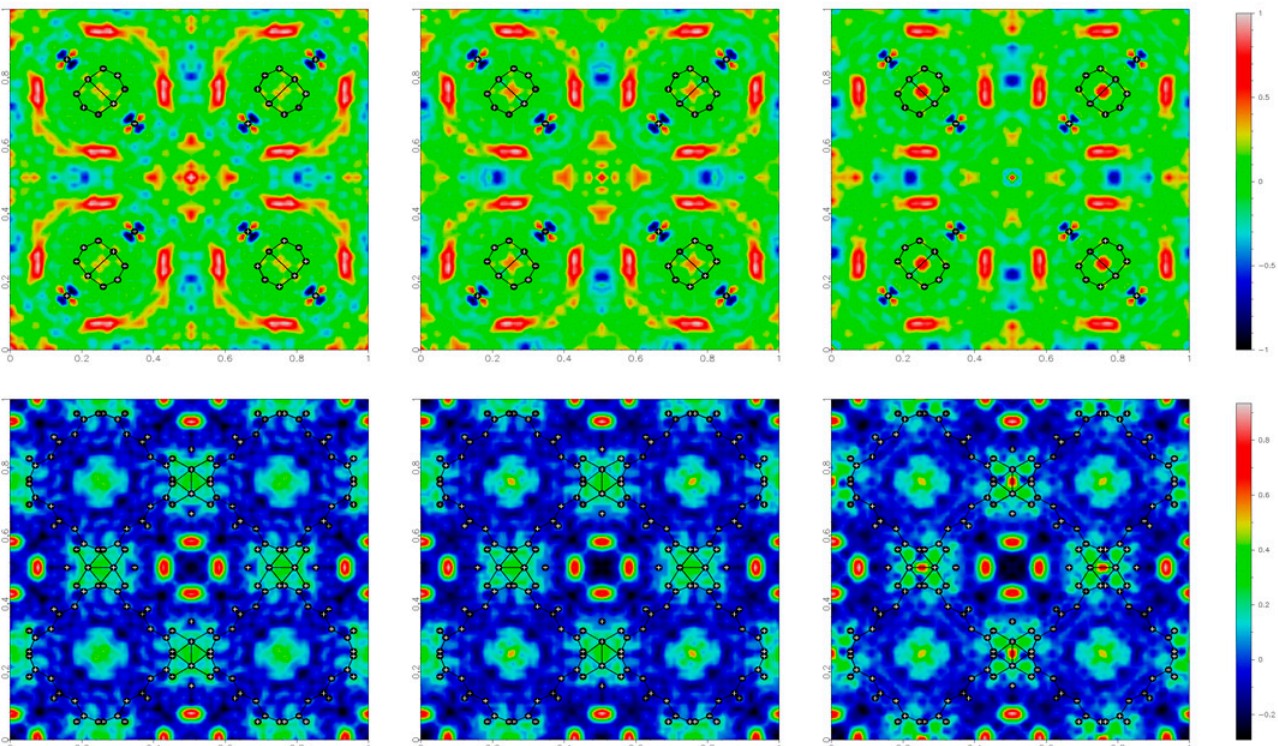

**Figure 4.** Residual electron density maps (obtained as Fourier summations of $F_o - F_c$) for HKUST-1, in the plane (x,y,0.5) on the top and (x,y,0.25) on the bottom, under different conditions: before, during and after the application of 1.5 kV potential, generating an electric field at the sample of ca. $1.5 \times 10^{-2}$ GV/m. The residual electron density values are color coded, as shown on the right side.

We also tested these samples under an electric field. From the previous experiments on hydrated samples, we noticed that HKUST-1 suffers from radiation damage, and therefore the radiation dose was reduced (also taking advantage of the higher scattering power of Br). Moreover, the diffraction experiments were carried out on two different single crystals: one used for experiment **2a** (without the field) and the other one for experiments **2b–2c** (during and after the application of the field). As for the hydrated forms, we calculated the Fourier difference maps (see Figure 5), including in the model the $CH_2Br_2$ molecules inside the tetrahedral cavities, whereas water molecules were not included in the refinement (apart from the oxygen atom of the water molecule coordinated to Cu).

Again, the application of the electric field only showed minor changes. First of all, no phase transition toward a polar space group was observed, caused, for example, by the reordering of the $CH_2Br_2$ molecules. There was only a slight decrease in the (isotropic) atomic displacement parameter of Br (though it was strongly correlated with the occupancy). There was an increase in electron density in the vicinity of the $CH_2Br_2$ molecules, also visible in the maps of Figure 5. These effects are partly maintained even after switching off the field.

### 3.2. Dielectric Behaviour of Protected HKUST-1

Adopting the strategy proposed by Gao et al. [9] (see also the Introduction), we protected HKUST-1 with various amines, as described in the experimental section (see also Table 1). After the reaction, the crystallinities of the materials were tested with X-ray powder diffraction (see Supplementary Material).

In order to test the efficiency in both enhancing the hydrophobic character of the MOF and preserving the same low-κ feature of the activated HKUST-1, we performed impedance spectroscopy on all the samples and determined the dielectric constant. κ is not typically adopted as an indicator of hydrophobicity. For this reason, we coupled the measurements

with gravimetric analysis in order to estimate the amount of water adsorbed within specific time ranges. In the next paragraph, we compare this technique with the more traditional water adsorption isotherms.

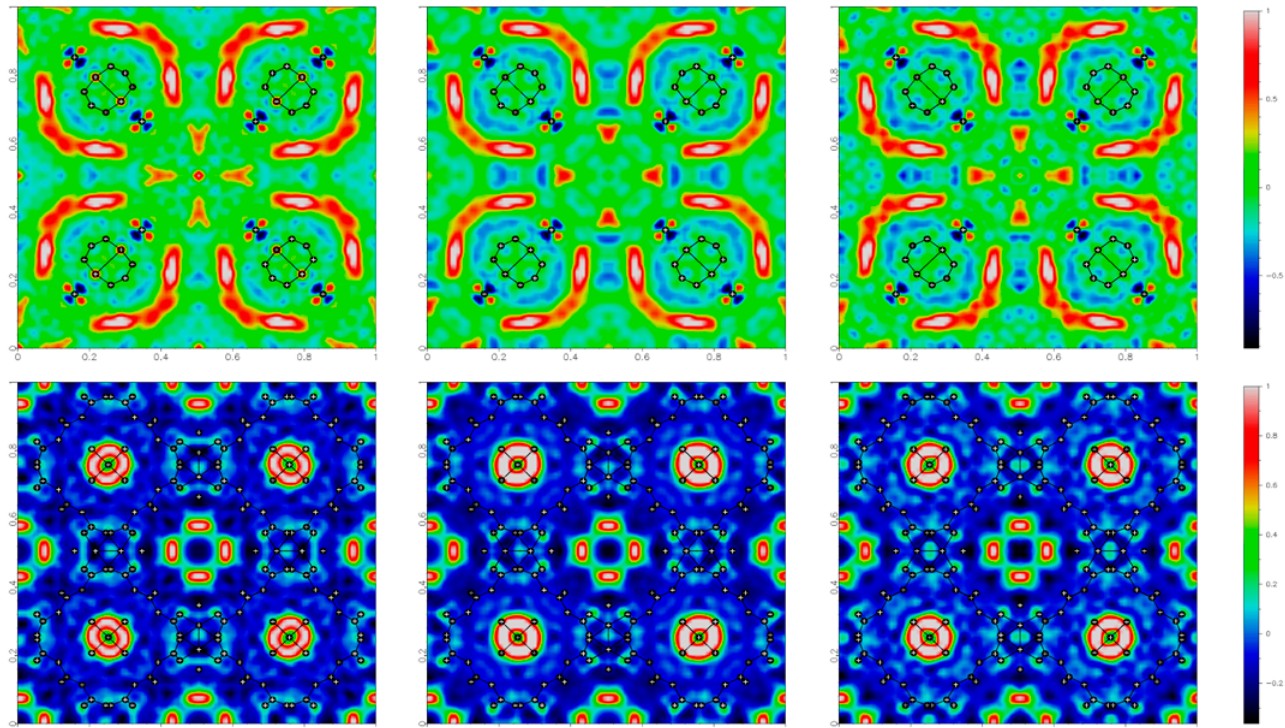

**Figure 5.** Residual electron density maps (obtained as Fourier summations of $F_o$–$F_c$) for HKUST-1@$CH_2Br_2$, in the plane (x,y,0.5) on the top and (x,y,0.25) on the bottom, under different conditions: before, during and after the application of 1.5 kV potential generating an electric field at the sample of ca. $1.5 \times 10^{-2}$ GV/m. The electron density values are color coded, as shown on the right side.

For the dielectric constant measurements, two parameters were analyzed: (a) κ at high frequency (1 MHz, the highest value reachable by our instrumentation) and (b) κ at low frequency (1 Hz). At high frequency, the dielectric constant simply depends on the polarizable electron density of the material, because the nuclear motion contribution is not yet activated. Therefore, the increase in κ (1 MHz) with respect to the as-activated material is directly proportional to the amount of matter (water) adsorbed per unit volume. At low frequency, instead, the nature of the binding of water molecules to the MOF may result in a higher or lower dielectric constant. In fact, as in the example reported by Scatena et al. [31], water molecules coordinated to Cu(II) ions do not contribute to the enhancement of κ at 1 Hz or lower, because they are quite tightly bound to the framework. Instead, molecules less tightly bound or free to move into the channel produce a significant alteration of κ at 1 Hz, but only a minimal alteration at 1 MHz. One could also consider the dielectric constant at lower frequencies, but the measurement would be significantly longer (the time being obviously inversely proportional to the frequency), without producing more information. However, a scan between 1 MHz and 1 Hz (typically repeated over 10 cycles) takes only few minutes and allows a series of measurements at regular time intervals after the material is exposed to a humid atmosphere. This enables precise adsorption kinetics curves to be derived.

In Figure 6, we report the correlation between the density of adsorbed water ($\rho H_2O$, based on gravimetry) and the two dielectric constant measurements (κ (1 MHz) and κ (1 Hz)) for unprotected HKUST-1 (top plot) and all the types of amine-protected HKUST-1 that we prepared. As anticipated, the correlation between $\rho H_2O$ and κ (1 MHz) is linear. Unprotected HKUST-1 after the activation gives the lowest κ (1 MHz) (1.78), whereas all

the amine-protected materials have higher baselines that obviously depend on the surface reactant itself (see below). The slope of κ (1 MHz) against $\rho H_2O$ is much larger for unprotected HKUST-1, compared to all amine-protected samples (only the 1naphtil-amine HKUST-1, 1NTA, has a similar trend). This means that the adsorbed water molecules may occupy different sites in unprotected and protected HKUST-1, therefore being polarizable in different ways. This is confirmed by the low-frequency dielectric constant, which grows exponentially in unprotected HKUST-1 (and similarly in 1NTA-protected HKUST-1), whereas it remains almost linear for all other amine-protected materials. The ideal saturation of all Cu(II) sites occurs for an adsorption of 0.078 g/cm$^3$ of $H_2O$.

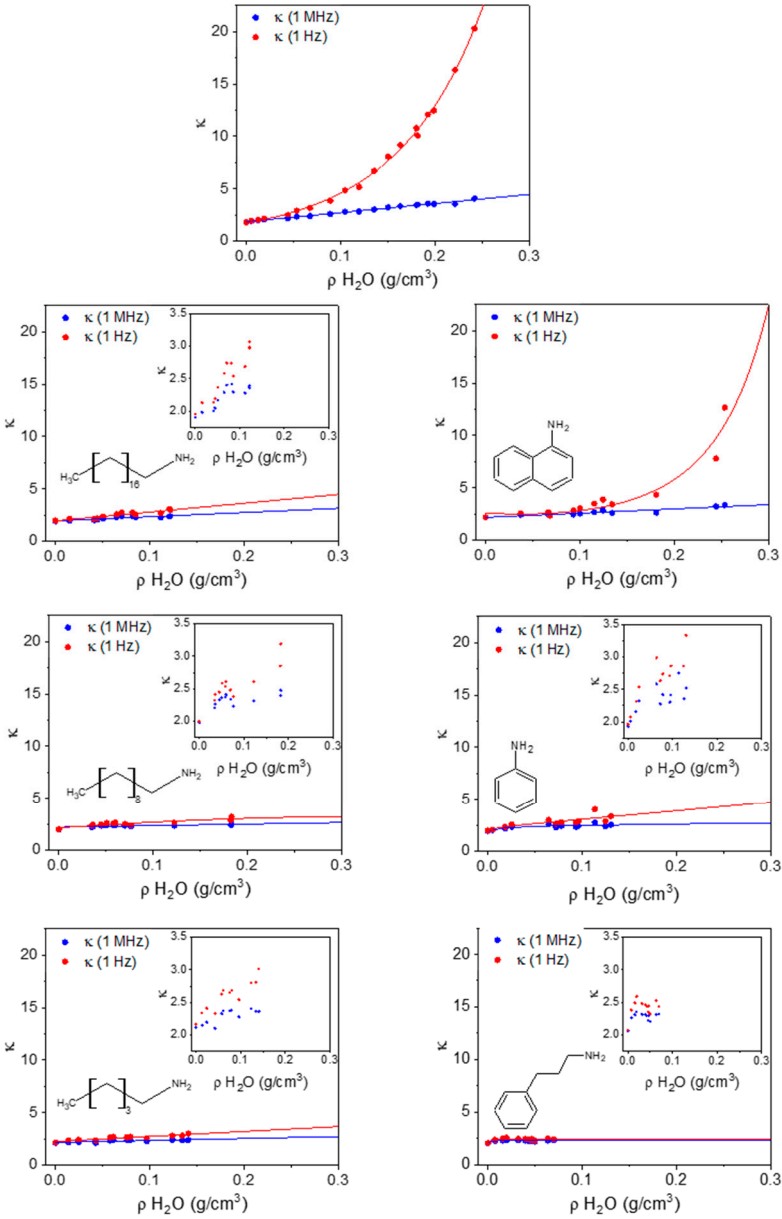

**Figure 6.** The dielectric constant at 1 MHz (blue) and 1 Hz (red) as a function of the mass density of adsorbed water in HKUST-1 and in all amine-protected species (**left** column shows alkyl amines; **right** column shows aromatic amines).

Because full occupancy of these sites does not occur before other water molecules start occupying the cavities, the deviation of the low-frequency κ from linearity (and from the high-frequency κ) begins for $\rho H_2O < 0.078$ g/cm$^3$ in unprotected HKUST-1.

In 3P1PA-protected HKUST-1 there is almost no difference between the low and high frequency, as it occurs for activated HKUST-1 when measured in a protected atmosphere and is therefore prevented from adsorbing water [27]. Nonetheless, the gravimetric analysis shows some water adsorption in 3P1PA; however, the protection annihilates all the negative effects of water sorption. The price to pay is a slightly larger $\kappa$ compared with activated HKUST-1. For all other amine-protected samples, apart from 1NTA, the low-frequency $\kappa$ is quite close to the high-frequency value, even though the saturation limit of Cu(II) sites is exceeded, and therefore water molecules surely bind also to other sites or are even free in the cavities of HKUST-1. Again, it seems that the effect of amine protection is that of severely reducing the mobility of water molecules inside the channels, even though the surface reactants were not able to prevent water sorption.

In Table 2, the calculated polarizabilities of the amines are reported, together with their expected contribution to the dielectric constant enhancement. In Figure 7, the molecular and atomic distributed polarizabilities are shown graphically. The $\Delta\kappa$ values are calculated assuming that the amines coordinate all the Cu(II) metal ions in the structure. This is obviously unlikely to occur, and therefore the experimentally measured increases in dielectric constant with respect to unprotected HKUST-1 are smaller. It is worth noting that octadecylamine, the most hindered amine, seems to occupy only 10% of the sites available, probably only those closer to the external surfaces of the crystallites. The smaller amylamine, on the other hand, almost saturates them, given that the observed $\Delta\kappa$ (1 MHz) is close to the expected value. For the aromatic amines, the dielectric constant increase is 50–65% of the theoretical amount.

**Table 2.** Isotropic polarizabilities $\alpha_{ISO}$ of the amines used for the protection of HKUST-1 and their contribution to the high-frequency dielectric constant of the material, assuming full saturation of the Cu(II) sites (the theoretical $\Delta\kappa$ is the expected increase with respect to the ideally empty $Cu_3(BTC)_2$). For sake of reference, the experimental $\Delta\kappa$ of the amine-protected HKUST-1 at 1 MHz is reported (the reference being the unprotected HKUST-1). The discrepancy between the expected $\Delta\kappa$ and the measured value indicates the degree of saturation of the Cu(II) sites.

| Protecting Amine | $\alpha_{ISO}$ (Bohr$^3$) | Theor. $\Delta\kappa$ | Expt. $\Delta\kappa$ (1 MHz) |
|---|---|---|---|
| Octadecylamine | 234.7 | 1.15 | 0.11 |
| Decylamine | 133.9 | 0.65 | 0.18 |
| Amylamine | 71.7 | 0.35 | 0.32 |
| 1-Naphthylamine | 125.6 | 0.61 | 0.39 |
| Aniline | 75.9 | 0.37 | 0.24 |
| 3-Phenyl-1-propylamine | 112.8 | 0.55 | 0.27 |

*3.3. Correlation between Isotherm Adsorption Curves and Dielectric Constant*

The results presented in the previous paragraph concern the dielectric behavior of protected HKUST-1 and clearly indicate that the surface reaction of amines significantly reduces the dielectric constant compared to that of unprotected HKUST-1, especially in the low-frequency regime, which is more sensitive to the mobility of polar guests.

The amount of water adsorbed (estimated from the weight increase of the measured pellets), can be verified with water adsorption isotherms. While adsorption isotherms of $N_2$ are used to measure the accessible surface area and pore volume, water adsorption measurements provide additional information due to the high polarity of the molecule and its hydrogen bond affinity, resulting in strong or weak interactions with the tested material. As reported by Canivet et al. [34], contradictory results for water adsorption isotherms are sometimes reported in the literature. HKUST-1 was found to be stable in water vapor by Alverez et al. [35], who reported a decreased capacity to adsorb water, but no worsening for other gases. Other authors, however, reported both a significant decrease in specific surface area and in water capacity after water adsorption. [36,37].

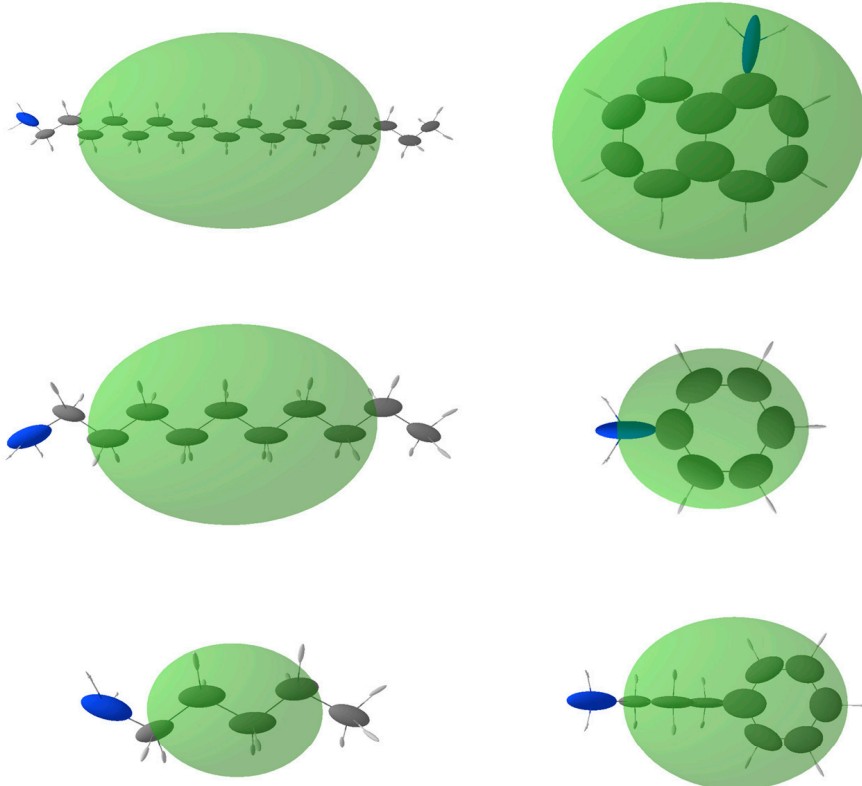

**Figure 7.** Molecular and distributed atomic polarizabilities calculated for all amines used in this study, using the PolaBer software [22]. Atomic polarizabilities are represented as atom-centered ellipsoids (colored according to the atom type: white for H, gray for C and blue for N), scaled to 0.2 Å$^{-2}$ for proper representation in the plot. Molecular polarizabilities are in transparent green, centered at the center of mass of the molecule and multiplied by a scale factor of 0.1 Å$^{-2}$.

From Figure 8 HKUST-1 shows a two-step adsorption process at lower pressure, indicating two energetically different mechanisms, which is in accord with the literature [32]. First, water molecules bind to the free copper sites. The second step in the adsorption isotherm is the filling of the large and small pores, which are less hydrophilic due to the absence of accessible metal sites and to the hydrophobic character of the benzene linker. A plateau is attained at about P/P$_0$ = 0.5, followed by an additional increase from P/P$_0$ = 0.8, possibly correlated with multilayer adsorption. Water adsorption isotherms were obtained also for DA-protected and 3P1PA-protected HKUST-1, i.e., the most efficient alkyl and aromatic amines, respectively. Compared to the unprotected HKUST-1, the two isotherms showed different shapes. The sigmoidal trend of the as-synthetized MOF was replaced by a logarithmic growth for both protected materials, confirming the enhanced hydrophobicity of the protected materials, in agreement with the impedance spectroscopy measurements. In particular, functionalization with 3-phenyl-1-propylamine seems to be the most promising approach for both techniques, confirming the utility of combining the two analyses in order to obtain a more complete result. In fact, the shape of the water isotherm indicates almost no interaction with water; only 2 mmol g$^{-1}$ at P/P$_0$ = 1.0 is adsorbed. Accordingly, the κ-value for 3P1PA is roughly stable over time (see Figure 9). The slightly worse performance of DA compared to 3P1PA emerges from the graphs shown. For example, the dielectric constant values at low frequency collected after several hours of exposure to air show a significant increase in κ, especially at low frequency (Δκ (1 Hz) = +23%).

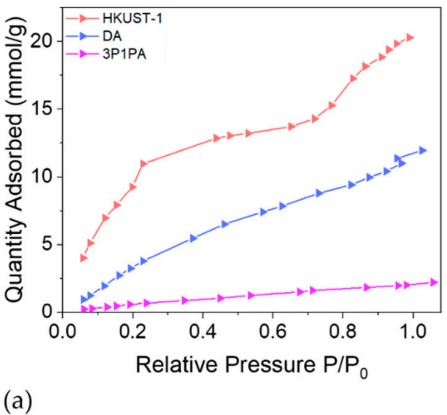
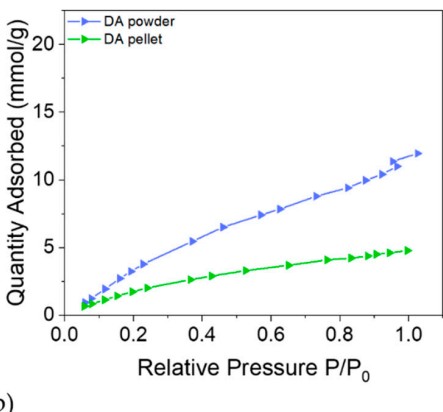

(a)                                                                                          (b)

**Figure 8.** (**a**) Quantity of water adsorbed per gram of unprotected HKUST-1, DA-protected and 3P1PA-protected substance as a function of the relative pressure. (**b**) Quantity of water adsorbed per gram of DA-protected HKUST-1 as function of relative pressure for a powder and a pressed pellet. Due to the smaller exposed surface area, water adsorption for pellets is significantly lower.

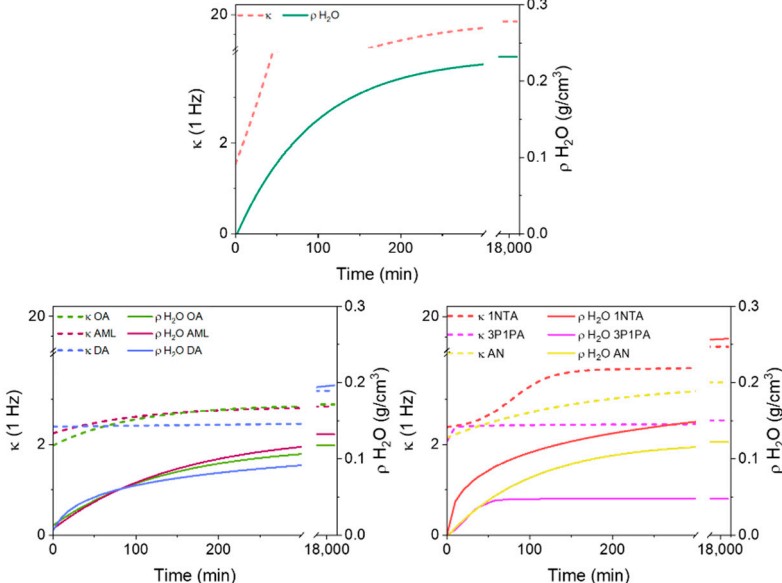

**Figure 9.** The dielectric constant at 1 Hz and the mass density of adsorbed water as a function of time. Top: HKUST-1; bottom left: alkyl amines; bottom right: aromatic amines.

In order to appreciate the effect of pelleting, another water isotherm was collected for a pellet of the DA sample and compared to the powder sample (See Figure 8b). As expected, due to the close packing of the particles (hence the smaller accessible surface), water adsorption for a pellet is significantly lower, while featuring a similar shape. In fact, in both cases, the trend is logarithmic, but at $P/P_0 = 1$ the amount of adsorbed water is 60% less for the pellet than for the powder.

## 4. Conclusions

The main hypothesis underlying the application of MOFs as low-$\kappa$ materials is the ideally empty space surrounded by an insulating framework that they can by definition guarantee. This feature reduces $\kappa$, as this is approximately the average between a vacuum and the hypothetically dense framework. Thus, the larger the proportion of empty space, the more efficient the material should be.

Unfortunately, hygroscopicity makes most highly porous MOFs extremely vulnerable, because they are easily and rapidly filled by the water molecules inevitably present in

the atmosphere with which the material is in contact. $H_2O$ prevents a low dielectric constant because it is strongly polar, highly polarizable and easily mobile inside the MOF channels. Thus, water increases both the polarizability of the material electron density and the electric-field-induced re-orientation and translation of molecular dipoles.

In this work, we investigated how one very important and often-studied MOF, namely HKUST-1 based on the $Cu_3(BTC)_2$ framework, can be protected, significantly increasing its hydrophobic behavior to overcome the main limitation regarding its use as a low-κ material for advanced microelectronics applications. Among the various remediations reported in the literature, we focused on a recently proposed methodology, i.e., a composite of the MOF and a surface-reacted amine [8]. For this purpose, we tested several aromatic and aliphatic (medium-long chain) amines, to check the overall success of the strategy and identify which amine displayed the most promising behavior.

The results presented and discussed in the previous sections enable some conclusions to be drawn:

- HKUST-1 is known to be a highly porous and hydrophilic material, and our experiments demonstrated that it is also stable in an electric field. In fact, no significant distortion of the framework was observed upon application of an external voltage. From another perspective, although the water guest molecules in the pores and channels do indeed react to the application of a field, they do not order significantly, to become more observable than in the absence of a field.

- The diffraction experiments on HKUST-1 after changing the guest from water to $CH_2Br_2$ revealed the differences between the two topological classes of pores: the tetrahedral (smaller) cavities trap $CH_2Br_2$, whereas the octahedral (larger) cavities cannot block them. This obviously implies that smaller guest molecules also experience different types of interactions with the framework, depending on the pores they enter. In these experiments, only binding to the framework was distinguishable (at Cu(II) sites, at the carboxylic groups of the linkers, or without any direct interaction with the framework).

- The tested amine surface reaction significantly improved the transformation of HKUST-1 (or in principle other MOFs) into a truly hydrophobic material, while maintaining its crystallinity. The main proofs of this statement are: (a) the reduced adsorption of water in the bulk, as proved by gravimetric and adsorption isotherm experiments; (b) the small values and the significant stability of the dielectric constant along the range of the scanned frequencies; and (c) the stability over time of the low dielectric conditions. The last point, however, requires further testing and probably also optimization of the fabrication techniques.

- The comparison between vapor adsorption isotherms and dielectric constant measurements indicated that a new perspective can be adopted when investigating the adsorption properties of MOFs. The measurement of the dielectric constant was quite rapid (although requiring a significant amount of material) and provided a response that agreed with the traditionally adopted adsorption isotherms. In particular, the protected MOFs (except for 1-NTA) revealed a single-stage mechanism, which is evident from the fact that κ (1 Hz) was quite similar to κ (1 MHz), growing linearly with time, rather than exponentially.

- Although in this study we explored only a limited number of amines, it seems evident that an alkyl chain improves the performance of the amine. Indeed, although 3P1PA (the most efficient) can be classified as an aromatic amine, it possesses a medium-length chain separating the aromatic ring from the amino group. On the other hand, amino groups directly linked to the aromatic ring do not seem to be so efficient. This is especially true for 1NTA (featuring a hindered aromatic system). Probably the combination of an aromatic ring and an alkyl chain is the best solution, because it combines the anchoring ability to the framework binding sites typical of flexible alkylic chains and the inherent hydrophobicity of aromatic rings.

Further work is needed to identify the most efficient amine (testing a larger group of amines). The best amine should be able to guarantee the highest performances and better stability of the material over time, which has not been investigated in this work.

A final remark should not escape the reader's attention. In this work we adopted impedance spectroscopy to semi-quantitatively assess the amount of water adsorbed by an MOF and correlate the observed signals with the type of guest adsorption into the pores. This technique could become complementary to other well-accessed methodologies for measuring water adsorption and could potentially be used also with other kinds of guests.

**Supplementary Materials:** The following supporting information can be downloaded at: https://www.mdpi.com/article/10.3390/chemistry4020041/s1: powder X-ray diffraction measurements of all the samples tested (Figure S1, as synthesized powders; Figure S2, as pellets obtained from compression of the powders); table of the single-crystal X-ray diffraction experiments (Table S1); cif files and checkcif reports for all single-crystal X-ray diffraction experiments (**1a**, **1b**, **1c**; **2a**, **2b**, **2c**).

**Author Contributions:** Conceptualization, P.M. and S.S.; methodology, P.M., S.S., N.C., V.C. and F.B.; software, P.M.; resources, P.M.; data curation, S.S., N.C. and F.B.; writing—original draft preparation, P.M.; writing—review and editing, P.M. and S.S.; funding acquisition, P.M. All authors have read and agreed to the published version of the manuscript.

**Funding:** This research was funded by the Italian MIUR (PhD grant for S.S.), the Polytechnic of Milan (Research Grant for P.M.) and the center for high-performance computing CINECA (project POLDIPOL). V.C. thanks the Italian MIUR for partial funding through the PRIN2017 project "Moscato no. 2017KKP5ZR) and the Università degli Studi di Milano (Transition grant PSR2015-1721VCOLO_01).

**Institutional Review Board Statement:** Not applicable.

**Informed Consent Statement:** Not applicable.

**Data Availability Statement:** Not applicable.

**Acknowledgments:** We thank Michael Lange for assisting during the experiments at SLS.

**Conflicts of Interest:** The authors declare no conflict of interest.

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
