# Peer review of "The Dielectric Behavior of Protected HKUST-1"

_chemistry, doi:10.3390/chemistry4020041_

Round 1

Reviewer 1 Report

The authors of the manuscript entitled: "The dielectric behaviour of protected HKUST-1" presented comprehensive studies of the sorption and dielectric characteristics of Cu3(BTC)2 MOF before and after protection with diverse amines. They proved that the application of amines is promising for reducing the inherent hygroscopic nature of MOF. Consequently, they bring us closer to the practical application of this material with a low dielectric constant. Moreover, they performed synchrotron X-ray diffraction studies in the intense electric field, which confirmed significant differences between structures with adsorbed water and dibromomethane. Finally, they demonstrated that impendence spectroscopy can provide complementary data about sorption inside MOF. These studies are a valuable contribution to scientists working on low dielectric constant materials and functional MOFs. Therefore, this paper is suitable for publication in Chemistry journal after minor corrections.

 1) Please refer to the research done by Tan et. al. in Adv. Mater. Interfaces 2020, 7, 2000408.

2) Correct the word “Tope” in the Figure 8 caption.

Author Response

We thank the reviewer for the comments. We have corrected the manuscript according to the minor remakrs of the reviewer

1) Please refer to the research done by Tan et. al. in Adv. Mater. Interfaces 20207, 2000408.

Cited in line 57.

2) Correct the word “Tope” in the Figure 8 caption.

Corrected

Reviewer 2 Report

The impedance spectroscopy was used to show the amount of water absorbed by a typical MOF, namely HKUST-1. The work found some correlation between the water adsorption and the dielectric constant. Furthermore, the stability of MOF upon a strong electric field has also been reported. This work is exciting for the community. However, there are also some weaknesses:

1) What about the dielectric loss in this work?

2) How to stabilize air humidity nearby ca. 60%? The experimental details could be supported;

3) The single crystal was used in X-ray diffraction measurements. However, section 2.1 did not indicate the information of the single crystal, such as size.

4) The discussion in Figure 6 is missing;

5) On Page11, Line398, It shows that “are even free in the cavities of HKUST-1”. What is the implication of this explanation? Why does the free H2O not affect the low-frequency κ here? However, on Page 9, it shows that “free to move into the channel produce  significant alteration of κ at 1 Hz”. 

7) The experimental section shows that the sample holder consists of a two brass parallel electrodes capacitor. So, did the conductive glue or electrode have been used as the electrode in the dielectric measurement? The information should be supported;

If all of these points could be addressed, this work could be re-considered.

Author Response

We thank the reviewer for the comments and the suggestions. 

We have modified the manuscript and teh SI in order to consider the reviewer's remarks. In particular

1) What about the dielectric loss in this work?

We added in the SI the tan(delta) plots, like the epsilon plots in the main text. These show that the dielectric loss increases for those species for which the dielectric constant increases, as a result of the adsorbed water.

2) How to stabilize air humidity nearby ca. 60%? The experimental details could be supported;

In lines 151-153 we added: “After the initial measurements at time t0, all the pellets were expose to a stable air humidity of ca. 60% (maintained by storing the samples in a closed box containing humidifier polymers and controlled through a hygrometer) ....”

3) The single crystal was used in X-ray diffraction measurements. However, section 2.1 did not indicate the information of the single crystal, such as size.

The information was actually reported in the cif files. We have now added the dimensions of the samples in lines 162-163)

4) The discussion in Figure 6 is missing;

A reference to Figure 6 was missing. Because the atomic and molecualr polarizabilities were discussed after Figure 7, we have exchanged Figures 6 and 7 and added a reference to Figure 7 (line 405). 

5) On Page11, Line398, It shows that “are even free in the cavities of HKUST-1”. What is the implication of this explanation? Why does the free H2O not affect the low-frequency κ here? However, on Page 9, it shows that “free to move into the channel produce significant alteration of κ at 1 Hz”. 

As explained in lines 399-401, the presence of amines remarkably reduce the effect of free H2O in the cavities at low frequency. Probably, due to weak interactions between amines and guest molecules. On page 9, the sentence is referred to water completely free to move in the pores, creating no interactions, as it happens in untreated HKUST-1.

7) The experimental section shows that the sample holder consists of a two brass parallel electrodes capacitor. So, did the conductive glue or electrode have been used as the electrode in the dielectric measurement? The information should be supported;

For the dielectric measurements no conductive glue was used, after realizing that the effect of the conducting glue was minimal (in previous experiments on HKUST-1 we used in fact the conducting glue). A sample is pressed as a pellet and placed between the two brass parallel electrodes, like a sandwich structure. Indeed, the parallel electrodes have been used as the electrode in the dielectric measurements.

Reviewer 3 Report

This manuscript by Sorbara et al. covers a topic of growing importance. Particularly considering importance of the niche (yet fast evolving) area of MOF based dielectrics, I find the manuscript (including excellent data quality) a timely addition on this subject. Therefore, I will be glad to support publication of a revised version. My comments are detailed as below:

1. The introduction section needs to include a concise discussion on hydrolysis of Cu–O bonds in the paddlewheels, especially considering their mechanistic investigations. As reference, the authors may see: J. Phys. Chem. C 2016, 120, 12879–12889; J. Phys. Chem. C 2020, 124, 1991–2001. Such a discussion may lead the authors to the concept of developing hemilabile MOFs, please see Nature Chemistry, 2018, 10, 1096–1102, DOI: 10.1038/s41557-018-0104-x.

2. Figure 7 needs a couple of changes: a) the inset figures need to use larger fonts for the X- and Y-axes to improve the readability; b) the inset figures' scales do not look right, e.g., the one at the bottom-right corner has all the datapoints crammed into a tiny section; c) Some of the top (X-) axis appears missing from the plots.

3. Figure 8, the top plot's X-axis needs a clear data separator mark "//" like the ones used in the bottom row plots.

Author Response

We thank the reviewer for the comments . we have modified the manuscript to respond to the remarks

1. The introduction section needs to include a concise discussion on hydrolysis of Cu–O bonds in the paddlewheels, especially considering their mechanistic investigations. As reference, the authors may see: J. Phys. Chem. C 2016, 120, 12879–12889; J. Phys. Chem. C 2020, 124, 1991–2001. Such a discussion may lead the authors to the concept of developing hemilabile MOFs, please see Nature Chemistry, 2018, 10, 1096–1102, DOI: 10.1038/s41557-018-0104-x.

We have modified the introduction and added the suggested references (lines 97-102)

2. Figure 7 needs a couple of changes: a) the inset figures need to use larger fonts for the X- and Y-axes to improve the readability; b) the inset figures' scales do not look right, e.g., the one at the bottom-right corner has all the datapoints crammed into a tiny section; c) Some of the top (X-) axis appears missing from the plots.

a) We have increased the inset figures’ fonts to improve the readability.

b) We have used the same scale in all the inset figures in order to highlight the different behaviour of protected-HKUST-1, e.g., the one at the bottom-right corner has all the datapoints crammed into a tiny section because of its major hydrophobicity in comparison to the others.

c) In HKUST-1 and 1NTA there was no need to plot the inset because of the rapid increase of kappa as a function of water adsorption. This avoids including a useless inset that may hamper a full picture for these two samples.

3. Figure 8, the top plot's X-axis needs a clear data separator mark "//" like the ones used in the bottom row plots.

A clearer data separator mark “//” is now used in the top plot’s X-axis